# Authors’ Protocol of Central Giant Cell Granuloma Effective Treatment in the Jawbone

**DOI:** 10.3390/cancers17213510

**Published:** 2025-10-31

**Authors:** Dominik Szczeciński, Patrycja Ujma, Katarzyna Radwańska, Piotr Szymor, Marcin Kozakiewicz

**Affiliations:** Department of Maxillofacial Surgery, Medical University of Lodz, 92-213 Lodz, Poland

**Keywords:** central giant cell granuloma, CGCG, jaws, children, treatment, surgery, radicalization, Carnoy solution, steroids, recurrence

## Abstract

**Simple Summary:**

Central giant cell granuloma of the jaw is a benign and rare disease (incidence of approximately 1–2 per 1,000,000 individuals per year). The solid tumor is a potentially aggressive bone lesion that can cause pain, facial deformity, tooth loss, and pathological fractures. Standard surgical treatment often results in significant tissue loss and functional impairment, especially in younger patients. This study presents a minimally invasive treatment protocol combining multiple weekly intralesional steroid injections with removal of residual tumor tissue and chemical cauterization with Carnoy’s solution to reduce recurrence. The approach allows preservation of jaw structure while effectively controlling tumor growth. Clinical and radiological follow-up demonstrated that the method is safe, well-tolerated, and can prevent recurrence when applied consistently, even in aggressive cases. These findings offer a practical strategy for clinicians, improving patient outcomes and supporting further research on conservative management of central giant cell granulomas.

**Abstract:**

Background: Central giant cell granuloma of the jaw is a benign but potentially aggressive lesion that can cause pain, facial deformity, tooth loss, and jaw destruction. Many treatment methods are described in the literature, but the less invasive ones are associated with a higher recurrence rate. For several decades, extensive bone resection procedures have been the most effective treatment to date. This study aimed to evaluate a minimally invasive treatment protocol combining multiple weekly intralesional steroid injections with surgical removal of residual tumor tissue and chemical cauterization using Carnoy’s solution. Methods: Thirteen patients with histologically confirmed central giant cell granulomas of the jaws were treated according to the protocol, including weekly triamcinolone injections and, when necessary, fenestration of the cortical bone to access residual lesions. Patients were monitored clinically and radiologically over six years, with reconstruction of bone defects using autogenous grafts and platelet-rich fibrin. Results: The treatment effectively reduced tumor size, restored cortical bone, and allowed preservation of jaw structure. Only one recurrence was observed, and complications were minor and transient. The protocol was equally effective for both aggressive and non-aggressive lesions, regardless of patient age or comorbidities. Conclusions: These findings suggest that combining pharmacological and surgical approaches with chemical cauterization provides a safe, effective, and tissue-preserving strategy for managing central giant cell granulomas, minimizing recurrence while reducing surgical morbidity.

## 1. Introduction

Pathologists describe the central giant cell granuloma (CGCG) of the jaw as a localized, benign (though sometimes aggressively behaving) osteolytic lesion of the jawbones, containing osteoclast-like giant cells in a vascular stroma. Epidemiology indicates that CGCG accounts for around 10% of benign gnathic tumors. It has an incidence of approximately 1–2 per 1,000,000 individuals per year, affecting mostly patients in the second and third decades of life. According to Choung et al., CGCG can be classified based on its clinical course and symptoms into a non-aggressive form and an aggressive form, the latter of which constitutes approximately 30% of these lesions [1,2,3]. The aggressive type of CGCG demonstrates pain, rapid rate of growth, significant swelling, often tooth root resorption and cortical perforation, as well as recurrences. CGCG can be a significant therapeutic challenge, especially in patients with its aggressive form. Though the lesion is histopathologically benign, the clinical course may be highly destructive. It can cause severe pain, significant bleeding and rapid bone destruction, leading to dentition loss, facial disfiguration, and in some cases, pathological bone fractures.

Various treatment methods have been proposed in the medical literature [4,5,6]:Surgical: Resection with a margin of unaffected tissues, either as a marginectomy or in more advanced cases, bone ostectomy;Simple curettage;Curettage followed by radicalization (ostectomy or the use of liquid nitrogen or Carnoy’s solution);Local pharmacological agent deponation, including steroids, calcitonin, interferon, imatinib, or denosumab.

The recurrence rate of the lesions, according to various authors, can range from 11% to 72%, depending on the chosen treatment method and the clinical form of the disease [1,7,8].

Although the primary treatment for CGCG is surgical resection of the affected bone, in recent years several studies have reported intralesional steroid injections (primarily triamcinolone) in various forms as a suitable treatment for CGCG. However, in many cases, patients required additional procedures after the completion of steroid therapy (mostly curettage of the lesions from the bone) [5,9,10,11]

The aforementioned factors, along with the relatively young population affected by the disease, encourage the search for the best treatment protocol that, on one hand, will maximally preserve the patient’s craniofacial tissues, while on the other hand, will minimize the risk of recurrence.

The objective of this study is to propose an efficient protocol for the management of CGCG, derived from our clinical experience.

## 2. Materials and Methods

The clinical material originates from the Department of Maxillofacial Surgery at the Medical University of Łódź, Poland, from 2015 to 2025, 13 patients were treated according to the protocol [Figure 1]—nine females and four males. The study was conducted according to the guidelines of the Declaration of Helsinki and approved by the bioethics committee at the Medical University of Łódź (numbers: RNN/48/15/KB 20/01/2015 and RNN/132/21/KE date: 11 May 2021).

The proposed treatment protocol [Figure 1] included patients in whom the presence of CGCG was confirmed by histopathological examination (patients diagnosed in our clinic or referred to the hospital outpatient clinic from other centers).

Inclusion criteria were as follows:At least one central giant cell granuloma of the jaw bones was confirmed in histopathological examination;Informed consent for treatment option;Attending regular check-up appointments.

Exclusion criteria were as follows:Absence of consent for the treatment method;Presence of brown tumor based on laboratory tests of parathyroid hormones and calcium-phosphate metabolism;Severe systemic conditions contraindicate surgical procedures (ASA physical status classification system category 3 and above).

Following the medical history, physical examination, biopsy and computed tomography (CT) scan, patients were directed to undergo laboratory blood tests (parathyroid hormone level, calcium, phosphates, vitamin D3, and alkaline phosphatase). Determining the normal levels of parathyroid hormone and calcium-phosphate metabolism allowed the exclusion of the presence of a brown tumor resulting from hyperparathyroidism, which is indistinguishable on histopathological examination from CGCG [12,13].

The first planned procedure involved removing the cortical portion of the jawbone that had formed over the tumor to create the so-called bone window, i.e., a fenestration. Creating a bone window allowed for the relatively easy insertion of a thick needle through the gingival/mucosal tissues to reach the actual CGCG mass.

The patients were administered injections on a weekly basis. The injection consisted of a mixture of trimacinolone acetonide 40 mg/mL (KENALOG^®^-40 injection, Bristol-Myers Squibb Company, Princeton, NJ 08543, USA) in a 1:1 ratio with 2% lidocaine [14,15]. The injection was administered intralesionally using a 1.2 mm (18 G) needle until the medication began to form the injection site.

The volume of the administered drug was usually around 2 cm^3^ until it flows back [Figure 2] at the needle insertion site or starts to form a blister of solution under the mucosa. The administered volume must be adjusted to the size of the tumor. In the initial stages of treatment, when the tumor is relatively less calcified or in the case of large tumors, a higher drug volume may be required.

Delivering injections within the tumor can be achieved through different approaches. This may be done by creating bone windows directly through the mucosa, as described above, or via a postoperative wound (e.g., following biopsy or tooth extraction). In all cases, the least invasive access method for the patient is always chosen to minimize discomfort and complications. In this protocol, injections were administered exclusively via the intraoral route through the oral mucosa. Injections were administered by the same surgeon. During therapy, if a cortical bone plate developed or if there were difficulties in performing the injection, additional imaging studies were conducted to locate any remaining CGCG foci. In patients whose therapy exceeded four months and in whom no clinically detectable changes in lesion calcification were observed, follow-up CBCT examinations were performed every 4–6 months. This approach allowed assessment of whether the response to corticosteroid therapy continued to progress.

If intrabony lesions persisted under peripheral bone repair and there were difficulties in reaching them with the needle, a new bone window was created [Figure 3] or the old one was widened (usually with a bone drill, e.g., for an osteosynthesis screw).

Computer tomography evaluation was performed using RadiAnt DICOM Viewer (Medixant. RadiAnt DICOM Viewer [Software]. version 2024.1 URL: www.radiantviewer.com; accessed on 1 July 2025). A lack of further response to the injections (as evident on the CT scans) was interpreted as a lack of CGCG calcification progression, indicating the need to qualify the patient for a procedure to remove the residual focus (or foci). The procedure consisted of curettage of the residual focus (utilizing bone curettes with sharp tips), chemical radicalization with modified Carnoy’s solution (composition: absolute alcohol 6 mL, chloroform 3 mL, glacial acetic acid 1 mL, ferric chloride 1 g) [4,16,17,18,19] is more secure and more conservative than a typical–classical surgical tumor removal. A seton was introduced into the cavity after the removal of the residual lesion, and it was soaked with Carnoy’s solution, which was left in the cavity for 3 min while the surrounding soft tissues were protected [Figure 4 and Figure 5]. After thoroughly irrigating the bone cavity with saline solution, the bone defect was reconstructed using an autogenous bone graft (chips harvested from the mandible or bone from the anterior iliac crest) mixed with platelet-rich fibrin (PRF).

Subsequently, patients were monitored clinically and radiologically according to the following schedule: every 6 months in the first year, and then once a year for 5 years. To reduce exposure to ionizing radiation, CBCT imaging was preferred.

In the retrospective study, based on medical records, a database was created for the statistical analysis of patient data, including age, gender, comorbidities, calcium-phosphate metabolism results, tumor size at the start of treatment, the number of injections administered, formation of an osteosclerotic rim, and the size of the residual focus [Table 1]. Analyses were conducted for both qualitative and quantitative characteristics.

Statistical analysis was performed in Statgraphics Centurion 18 (Statgraphics Technologies Inc., The Plains City, Warrenton, VA, USA www.statgraphics.com accessed on 28 September 2025). The ANOVA or Kruskal–Wallis test was applied for between-design comparisons. Independent χ^2^ tests were used to test the categorical variables. The difference between the two quantitative variables was assessed by Student *t*-test or Mann–Whitney–Wilcoxon U test. A *p*-value less than 0.05 was considered statistically significant.

## 3. Results

### 3.1. General Characteristics

The patients’ age range in the group was from 8 to 61 years [Table 1]. One patient had two metachronously occurring foci of CGCG in the right and left maxilla. Only two patients were affected by CGCG located in the maxilla, while the remainder had it in the mandible. Nine lesions were classified, based on clinical and radiological symptoms, as the aggressive form of CGCG. A typical case is presented in Figure 6, Figure 7, Figure 8, Figure 9 and Figure 10.

The relationships between the characteristics studied in the presented cohort are shown in Table 2 below.

### 3.2. Injections

Based on statistical analysis, no correlation was found between the size of the primary lesion and the patients’ serum PTH levels. Furthermore, no association was observed between the response to triamcinolone injections and serum PTH levels. This confirms that CGCG is not a hormone-dependent lesion, in contrast to a brown tumor.

Therefore, parathyroid disorders should be excluded at the beginning of treatment. A higher frequency of occurrence was observed in women (*p* < 0.05), which is consistent with findings reported in the literature [7]. However, the therapy was equally effective for both males and females, regardless of age, and for both clinical forms of CGCG (aggressive and non-aggressive). No influence of comorbidities on the size of the residual lesion was found.

A clinically observed association between tumor reduction and the number of injections (on the borderline of significance, *p* < 0.0775, correlation coefficient = 0.66). It should be noted that this pharmacologically induced healing is peripheral: it leads to the relatively rapid regeneration of cortical bone around the tumor. This makes it difficult to proceed with the second step of the treatment protocol [Figure 11 and Figure 12]. Sometimes, it is necessary to repeat fenestration three times.

### 3.3. Osteosclerotic Rim (Peripheral Corticalization)

The boundary between the soft tissues and the tumor (the so-called osteosclerotic rim) after the completion of injections was present regardless of the patient’s gender, comorbidities, clinical type of CGCG, location, or whether the tumor was sharply demarcated from the surrounding tissues before the treatment [Figure 13]. The occurrence of corticalization should not, however, warrant the discontinuation of injection therapy. There is no correlation with recurrence, as corticalization alone does not prevent further development of CGCG.

### 3.4. Complications During Treatment

The complications observed during the injections were local in nature. The most common complication was minor bleeding from the injection site; in one patient with a maxillary tumor at the alveolar–nasal junction, bleeding from the nasal cavity was also observed. Simple measures, such as pressure with a gauze pad or compression of the nasal wings, were sufficient to control the bleeding. In cases of serial injections, signs of local inflammation in the form of purulent discharge around the tumor were occasionally observed (in four patients, most likely due to infection of a hematoma at the injection site), which were managed with a short course of empirical antibiotic therapy (amoxicillin administered orally).

Two patients experienced transient paresthesia affecting branches of the trigeminal nerve, which resolved following B-vitamin supplementation.

These complications should be considered in the informed consent obtained from patients who decide to undergo treatment of CGCG with steroid injections.

### 3.5. Recurrence

During the observation period, only one recurrence was observed—in a patient with CGCG of the mandibular body [Figure 14]. The recurrence occurred 12 months after the primary procedure. In patients with large primary foci, a recurrence of the disease should be expected (*p* < 0.005). The size of the residual lesion and the occurrence of recurrence do not depend on the number of injections.

However, the larger size of the primary lesion affects the risk of recurrence. This study indicates a relationship between recurrence and both the size of the primary lesion (*p* < 0.005) and the size of the residual lesion (*p* < 0.05). A good response to steroid injections does not exclude the possibility of recurrence and requires continued treatment.

No correlation was found between the studied parameters and the presence of adverse events during treatment. However, larger residual lesions are associated with an increased risk of paresthesia in the area innervated by the trigeminal nerve (*p* < 0.05). In our study, complications tended to subside over time.

In the single patient with recurrence, we found that the only factor significantly associated with recurrence (*p* < 0.05) was a low calcium level. Therefore, in cases of hypocalcemia, special attention should be given to detailed analysis of radiological studies and to ensuring patient compliance with follow-up visits.

## 4. Discussion

Central giant cell granuloma (CGCG) of the jaws remains a significant challenge for the oral and maxillofacial surgeon. This is reflected in the continuous emergence of scientific reports proposing various treatment modalities. Despite being regarded as the gold standard, surgical resection is associated with considerable morbidity, including facial deformity, malocclusion, and paresthesia resulting from the removal of branches of the trigeminal nerve [1,7,8,20]. Alternative therapeutic strategies, such as calcitonin administration, intralesional steroid injections, or denosumab therapy, despite requiring prolonged treatment and a higher degree of patient compliance due to their less invasive nature, represent attractive options not only for clinicians but also for patients [2,6,7,14,15,21,22,23,24,25]. In our series, all patients responded positively to the opportunity of undergoing a more conservative surgical approach, despite the necessity of multiple weekly outpatient visits and repeated injections.

Among pharmacologically based alternatives, other authors have proposed the use of monoclonal antibodies against RANKL, namely denosumab. This agent plays an important role in oncologic therapy and the management of osteoporosis. However, its well-documented association with medication-related osteonecrosis of the jaws (MRONJ) [26] compels us to favor therapeutic agents with lower selectivity that, unlike antiresorptives, do not directly increase the risk of MRONJ in the management of CGCG of the jaws. Moreover, systemic adverse effects of denosumab, even when administered intralesionally, have been reported in the literature [27,28,29]. In our material, consistent with other published series, we did not observe systemic complications associated with intralesional steroid injections.

An essential element of our protocol is the removal of residual lesions. Histopathological examination of postoperative specimens revealed persistent CGCG fragments in some patients. This procedure not only allows for remodeling of the frequently expanded bone but also eliminates residual tumor tissue that, without further injections, could lead to recurrence. Some publications suggest that intralesional corticosteroid injections alone are effective; however, recurrences were reported in many cases, and the overall assessment of treatment efficacy remained inconclusive [5,8,9,30]

Reconstruction of the jaws after bone resection is a major surgical challenge, often requiring staged, time-consuming, and costly procedures. In our experience, the use of modified Carnoy’s solution facilitates access to hidden recesses following curettage of residual lesions. Tumor calcification after injections often leaves multiple bony septa, and attempting to remove them by peripheral ostectomy to reach every crevice would result in greater loss of valuable bone [4,31,32]. Carnoy’s solution has been used in medicine for nearly a century. It has been used as an adjunct to chemical curettage following the removal of jawbone tumors, such as odontogenic keratocyst, unicystic ameloblastoma, juvenile ossifying fibroma, dentigerous cyst, and others. Despite the potential risk of injury to the inferior alveolar neurovascular bundle or damage to the maxillary sinus mucosa, it remains a valuable adjunctive tool for the surgeon. Due to the potentially harmful effects of one of its components—chloroform—some authors have proposed the use of a modified formulation without chloroform. However, available studies provide inconsistent evidence regarding the efficacy of the chloroform-free formulation [19,33,34,35].

The optimal follow-up period for patients treated for CGCG remains a matter of debate. Different authors report observation intervals ranging from several months to 10 years. Based on our experience, the 6-year period of radiological follow-up adopted in our study appears sufficient to detect potential recurrences safely [8,11,30,36,37,38].

The limitations of the present study include the relatively small cohort size and the heterogeneity of patients presenting with lesions of varying sizes, multiplicity, and clinical behavior (aggressive versus non-aggressive). Similar methodological challenges are encountered by other research groups investigating this condition [5,8], which is a direct consequence of its low incidence.

## 5. Conclusions

The proposed protocol is an effective tool in the medical fight against central giant cell granuloma of both types. Our approach offers a safer and more conservative alternative to conventional surgical tumor removal.

However, the large volume of the primary lesion and low calcium level affect the risk of recurrence. A good response to steroid injections does not exclude the possibility of recurrence and requires continued follow-up. No relation was found between the studied parameters and the presence of adverse events during treatment. No recurrence was observed after the proposed observation period.

## Figures and Tables

**Figure 1 cancers-17-03510-f001:**
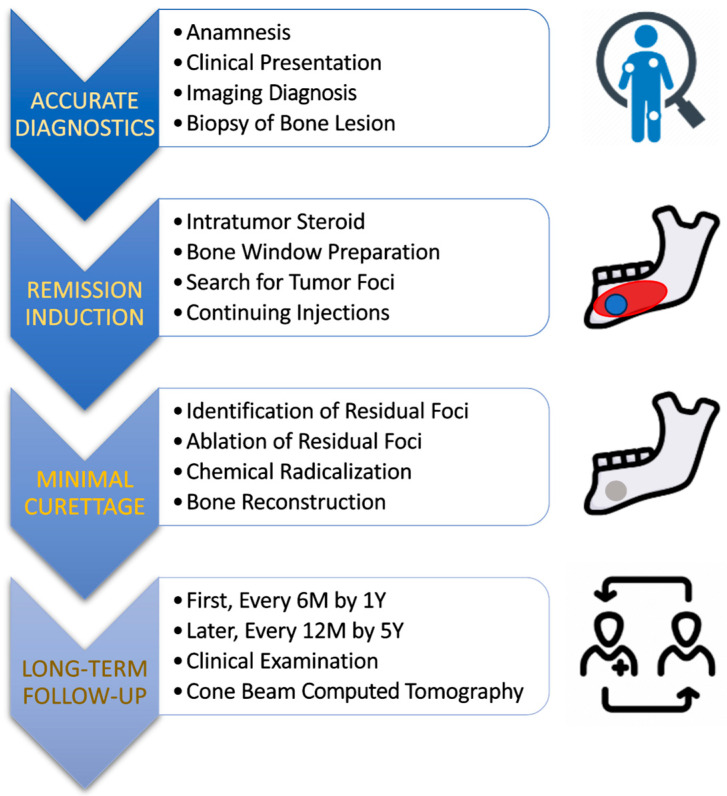
Proposed protocol to make CGCG treatment more effective and less invasive. The main factors leading to such a clinical goal are multiple intratumoral steroid injections and the combination of residual tumor removal in the remission phase with the application of chemical radicalization with Carnoy’s solution (composition: absolute alcohol 6 mL, chloroform 3 mL, glacial acetic acid 1 mL, and ferric chloride 1 g).

**Figure 2 cancers-17-03510-f002:**
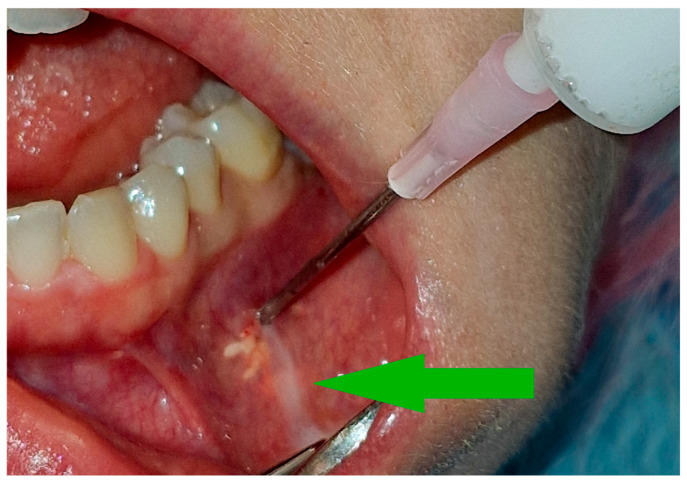
Therapeutic agent leakage through the needle insertion site is a signal to stop administering the medication. The white streak of the solution is indicated by the green arrow.

**Figure 3 cancers-17-03510-f003:**
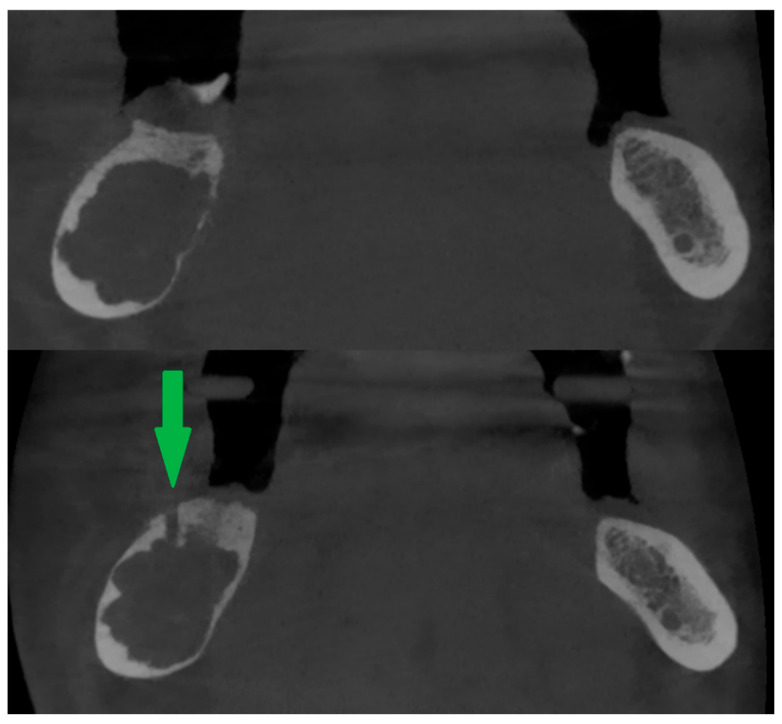
Peripheral calcification of the lesion (**top**). After creating a new bone window using a drill (**bottom**). Green arrow shows window in cortical bone.

**Figure 4 cancers-17-03510-f004:**
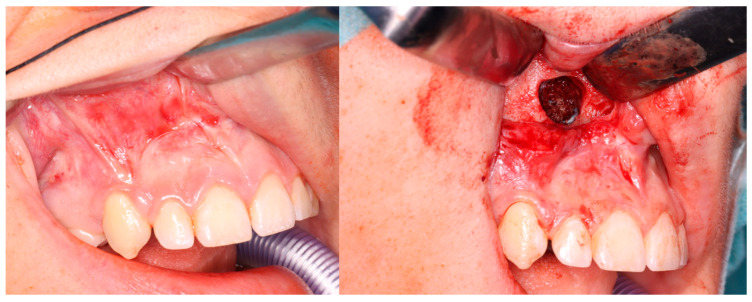
Intraoperative images. (**Left**): Patient prior to removal of the residual lesion in the anterior region of the right maxilla. (**Right**): Mucoperiosteal flap elevated, lesion curetted. A gauze seton soaked in Carnoy’s solution is placed in the bone defect (brown discoloration visible due to the fixing effect).

**Figure 5 cancers-17-03510-f005:**
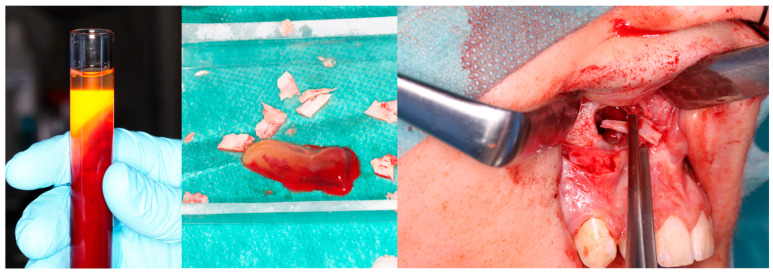
Same patient as shown in Figure 4. Peripheral venous blood was collected and centrifuged to obtain platelet-rich fibrin (PRF). Autogenous bone was mixed with PRF and placed into the defect following lesion removal.

**Figure 6 cancers-17-03510-f006:**
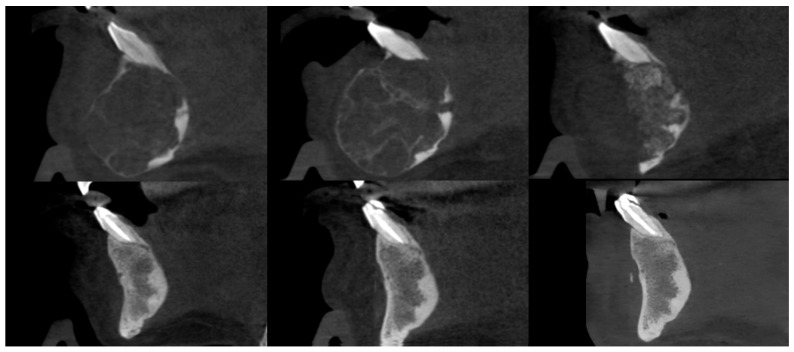
Female patient with a central giant cell granuloma (CGCG) of the anterior mandible. CBCT sagittal sections, from top to bottom and left to right: prior to injections, after injections, postoperative after residual lesion removal, 1-year follow-up, 2-year follow-up, and 3-year follow-up.

**Figure 7 cancers-17-03510-f007:**
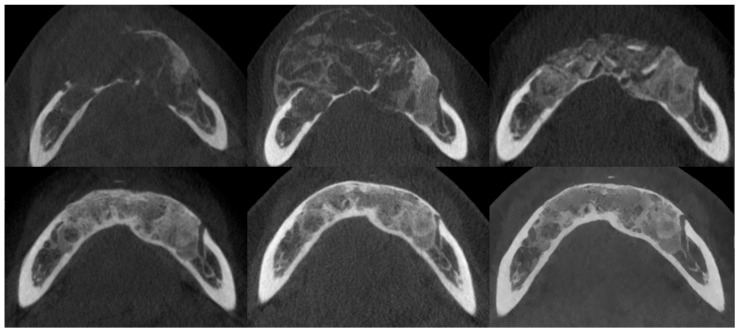
The same patient as shown in Figure 6. CBCT axial sections, from top to bottom and left to right: prior to injections, after injections, postoperative after residual lesion removal, 1-year follow-up, 2-year follow-up, and 3-year follow-up.

**Figure 8 cancers-17-03510-f008:**
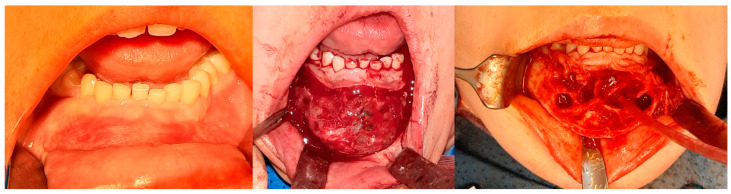
The same patient as shown in Figure 6 and Figure 7. Intraoperative photographs, from left to right: Before tissue incision; after elevation of the mucoperiosteal flap—showing expansion of the mandibular bone with a rebuilt cortical layer; after removal of the residual lesion, with visible bony septa.

**Figure 9 cancers-17-03510-f009:**
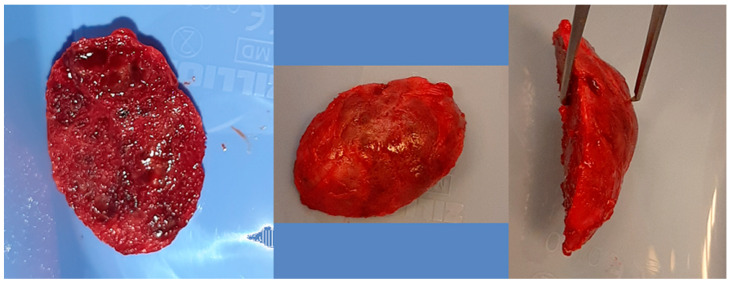
The same patient as shown in Figure 6, Figure 7 and Figure 8. Intraoperative photographs of the excised chin fragment and resection of excessive bone volume (the patient developed prognathism due to tumor growth); then, genioplasty was also required during the procedure.

**Figure 10 cancers-17-03510-f010:**
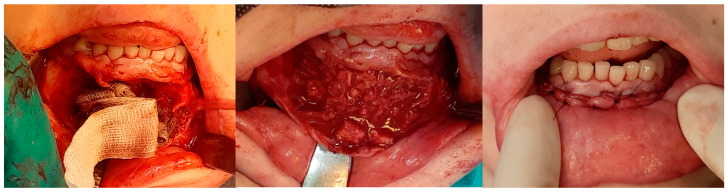
The same patient as shown in Figure 6, Figure 7, Figure 8 and Figure 9. Intraoperative view showing a Carnoy’s solution–soaked seton placed in the defect after CGCG removal, bone graft, and the surgically closed wound.

**Figure 11 cancers-17-03510-f011:**
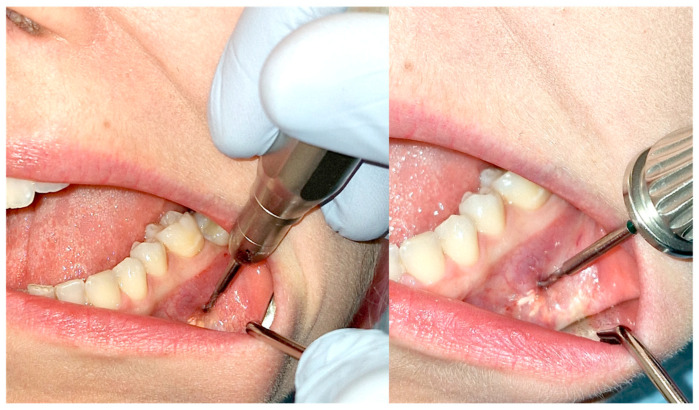
Repeat fenestration. On the (**left**), using a surgical handpiece; on the (**right**), with a drill attached to a screwdriver used for osteosynthesis.

**Figure 12 cancers-17-03510-f012:**
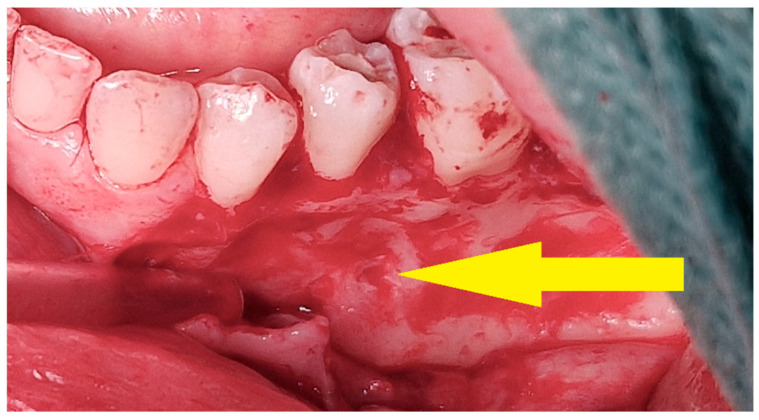
Intraoperative view—surgical removal of a residual lesion. After elevation of the mucoperiosteal flap, a bony depression is visible at the site of multiple intralesional injections (yellow arrow).

**Figure 13 cancers-17-03510-f013:**
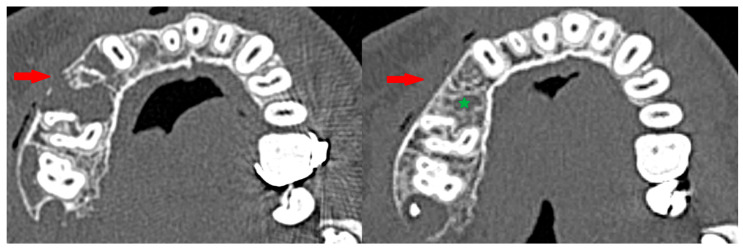
Peripheral corticalization of bone fenestration (red arrow); after corticalization, there is still a residual tumor (green star).

**Figure 14 cancers-17-03510-f014:**
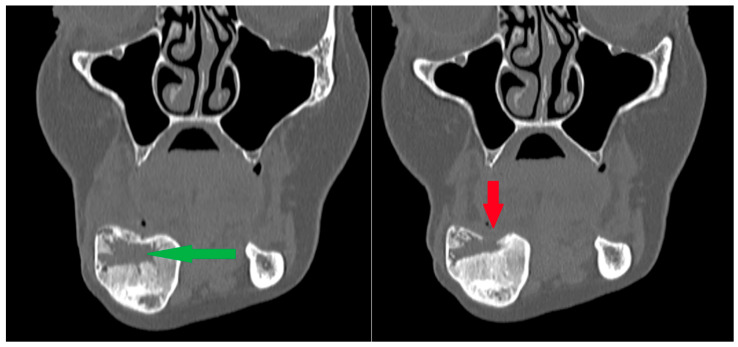
Patient following removal of a residual lesion and reconstruction of an extensive defect with a bone graft. (**Left**): The green arrow indicates the center of the defect during the healing process after grafting, with visible bone remodeling; the outer cortical layers remain preserved. (**Right**): Subsequent follow-up radiograph showing arrested healing and cortical bone destruction (red arrow). Recurrence was suspected and later confirmed by surgical excision of the lesion. Note: a second radiolucency is located near the lower border of the mandibular body—it spontaneously calcified during the follow-up period.

**Table 1 cancers-17-03510-t001:** General information about the patient group.

*n*	Sex	Age	Comorbidity	Number of Injections	Imaging Follow-Up	Clinical Follow-Up [Months]	Jaw Localization	CGCG Type
1	Female	61	DM	25	Healthy	68	Mandible	Non-Aggressive
2	Female	25	IBS	12	Healthy	71	Mandible	Aggressive
3	Male	27	AF	7	Healthy	55	Maxilla	Aggressive
4	Male	8	No	65	Healthy	34	Maxilla	Aggressive
5	Male	27	No	1	Healthy	50	Mandible	Aggressive
6	Female	18	No	8	Healthy	107	Mandible	Aggressive
7	Female	41	No	6	Healthy	115	Mandible	Aggressive
8	Female	28	No	3	Healthy	94	Mandible	Non-Aggressive
9	Female	46	HTN	5	Healthy	101	Mandible	Non-Aggressive
10	Male	31	No	36	Healthy	31	Mandible	Aggressive
11	Female	61	HTN	26	Recurrence	32	Mandible	Aggressive
12	Male	31	AF	12	Healthy	34	Maxilla	Aggressive
13	Female	33	No	2	Healthy	20	Mandible	Non-Aggressive

AF—atrial fibrillation; HTN—arterial hypertension; IBS—irritable bowel syndrome; DM—diabetes mellitus.

**Table 2 cancers-17-03510-t002:** The table reports *p*-values corresponding to the statistical significance of the analyzed qualitative and quantitative variables. A hyphen (“-”) indicates instances in which significance testing was not undertaken.

Variables of the Studied Group	Lesion Volume Before Injection	Lesion Volume After Injection	Percentage of Primary Lesion After Steroid Injections	Imaging Follow-Up (Healthy/Recurrence)	Lesion Perimeter After Injection
Age	0.11	0.25	0.30	-	-
PTH [pmol/L]	0.31	0.19	0.32	0.19	-
Gender	0.028	-	0.81	-	0.2
Comorbidity	0.63	-	0.3	-	0.68
Jaw localization	0.31	-	-	1.0	0.55
CGCG type	0.52	-	0.17	1.0	0.73
Side effects of therapy	0.22	-	0.0299 *	0.72	-
Calcium [mmol/L]	-	-	0.21	0.0075 *	-
Phosphates [mmol/L]	-	-	0.27	0.41	-
ALP [U/L]	-	-	0.0495 *	0.93	-
Number of injections	-	-	0.24	0.7	-
Imaging follow-up (healthy/recurrence)	0.0022 *	0.0137 *	0.45	-	1.0
Clinical Follow-up [months]	-	-	-	0.28	-

* Statistical significance.

## Data Availability

The data on which this study is based will be made available upon request at https://www.researchgate.net/profile/Marcin-Kozakiewicz (accessed on 24 February 2024).

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
