# Peer review of "Authors’ Protocol of Central Giant Cell Granuloma Effective Treatment in the Jawbone"

_cancers, 2025, doi:10.3390/cancers17213510_

Round 1
Reviewer 1 Report
Comments and Suggestions for Authors
I applaud the design and patient follow up. There are good thoughts in the design and use of operative intervention to assist with the clinical injections.
A few thought come to mind.
There is mention that cone beam is used for follow up owing to less radiation but perhaps an OPG could be used as well and is more readily available and has less radiation.
Also, Carnoy solution is historically shown to be an effective treatment in other pathology but there is concern over the carcinogenic effect of chloroform. Currently Carnoy solution is not approved by the US FDA; thus, there should be some comment as to the potential use of modified Carnoy solution.
Author Response
Dear Reviewer,
Thank you very much for your valuable review and for taking the time to provide thoughtful comments on our manuscript.
In response to your questions and remarks:
- Regarding our preference for CBCT:
- In our center, CBCT has long been considered the standard imaging modality for postoperative and follow-up evaluation of the jaws.
- The radiation dose to the patient is practically comparable, whereas the amount of diagnostic information obtained from CBCT is, in our opinion, incomparably greater than from OPG. According to the manufacturer’s data, the effective dose for OPG is approximately 52 µSv (equivalent to 6 days of background radiation), while for 16 × 10 cm CBCT it is around 136 µSv (16 days of background radiation). We consider this to be an excellent balance between the potential cost of radiation exposure and the diagnostic value gained.
- Due to differences in the training of radiology technicians, the use of various OPG machines, and patient head positioning, we find it difficult to obtain reproducible projections with OPG. These subtle differences are clinically significant when monitoring bone healing, where imaging consistency is crucial.
- Regarding the use of Carnoy’s solution:
Carnoy’s solution has been routinely used in our department for many years for chemical curettage following the removal of odontogenic tumors. Since we practice in Europe, the FDA recommendations do not have legal force in our jurisdiction. There are currently no regulations prohibiting the use of chloroform in medical applications. Its use in our center has been approved by the institutional bioethics committee, and all patients provide written informed consent.
Even if the carcinogenic potential of chloroform in humans were to be confirmed, our patients are clinically and radiologically monitored, and they are informed that any new or concerning symptoms in the treated area should prompt immediate medical consultation.
Several studies have suggested the superiority of Carnoy’s solution containing chloroform compared to the modified formulation without chloroform, for example:
- Dashow, J.E. Significantly Decreased Recurrence Rates in Keratocystic Odontogenic Tumor With Simple Enucleation and Curettage Using Carnoy's Versus Modified Carnoy's Solution. J Oral Maxillofac Surg. 2015. doi:10.1016/j.joms.2015.05.005
Other studies, however, have found no significant difference between the two formulations:
- Janas-Naze, A.; Zhang, W.; Szuta, M. Modified Carnoy’s Versus Carnoy’s Solution in the Management of Odontogenic Keratocysts—A Single Center Experience. J Clin Med. 2023, 12.
Interestingly, even after the restriction of Carnoy’s solution use in the United States, oral and maxillofacial surgeons continued to use the formulation containing chloroform:
- Ecker, J.; Horst, R.T.; Koslovsky, D. Current Role of Carnoy’s Solution in Treating Keratocystic Odontogenic Tumors. J Oral Maxillofac Surg. 2016; 74: 278–282.
We hope this explanation clarifies our rationale and methodology. Once again, we sincerely thank you for your insightful and constructive feedback, which we believe contributes to improving the quality of our work.
Reviewer 2 Report
Comments and Suggestions for Authors
hello
thank you very much for this interesting paper
authors presented their own retrospective paper about cgcg, a bone entity in the jaw bones
1.
title is ok
because authors have just 12 cases, I strongly suggest to change the title to - preliminary authors report - or- own obserwation based on 12 cases - or authors protocol on - please select the most appropriate one to highlight just those 12 cases and authors own retrospective results in this study
presented summary and abstract are sufficient - clear aim is set
used key words are fine, nothing more to change
abstract is OK, title needs a change
also - there is tvelve patients mentioned in abstract while the table1 has 13? please corect this major mistake in this retrospective patient database
2.
introduction is very short with good references
while authors did not listed also bone ostectomies, resection, marginectomies and other approaches to treat those lesions?
the treatment modalities are poorly presented
authors did not present with what cgcg should be differentiated in the terms od diagnostics and treatement
aim if study should be changed to either preliminary report or a report based on own approach and six years follow up? - why the full aim is not stated correctly?
3.
material and methods
bioethical protocol is presented
figure 1 lacks some explanation - like the timing of each step - does each step take the same time or is it case-dependent?
the inclusion and exclusion criteria - were there any age barier related or patients some other diseases that could affect the inclusion in the study? for example secondary tumors outside of the jaw bones? did authors only focused on jaw bone relations or where there any other relations?
are maxillary/mandible lesions treated in the same time frame and procedure?
how exactly CBCT helped authors to administer the solution to the desired spot?
the methods of delivering the solution should be listed in a table, to make it more easy to understand possible approaches and their good and weak sides
did athors only used carnoy and curretage? what about bone ostectomies or usage of alternative approaches?
was autologous bone used alone with prf or perhaps combined with some collagen membranes aswell?
was the cgcg tumor injections done by the same person and same technique? what was the standarisation for this approach? or perhaps there is noone existing and its all case-dependant?
table 1- what were the co-morbidities? did the affect on the results? list them in abbreviiations under table
I would change table gender, to sex
4
results
presented figures are very nice with a good scientific value
presented results are quite ok as for this preliminary study
table 2 - how was the lesion volume measured? was it done on ct or cbct and how?
table 2 somehow is not clear to fully understand- whats the correlation between cgcg type and lesion volume before injection?
the fenestration method - why it was not described earlier in the methods? please improve methods section and perhaps add a table or a paragraph describing possible surgical steps in chronological manner
should the size of the cgcg tumor and its location affect the numer of injections during one single visit, or perhaps each tumor had just one single injection?
this sentence should be removed: - t—surgeon cannot predict in advance which patients will experience complications - this is too obvious….
authors should highlight that their own approach is more secure and more conservative than a typical - classical surgical tumor removal
5
discussion and conclusions;
both should be improved and correlated authors approach with carnoy+curretage with other possible treatement outcomes
authors also should wrote about the role of cbct and possible other diagnostic methods that should be used and why
authors did not wrote if the biopsy was taken from the supericial or deep part of the lesions - was it one or more sample tissues, to evaluated them histopathologically and confirm the final diagnosis
6
used references are good
dear authors, thank you very much for this valuable study
based on the following comment, please improve the paper to make it more suitable for Cancers
current format has some weak spots
please improve the paper, it has a very good potential
thank you, with kind regards
Reviewer 3 Report
Comments and Suggestions for Authors
1, Fenestration surgery is relatively easy to perform and allows for gradual decompression of the cystic cavity. However, it often requires repeated aspirations and frequent follow-up visits, which may cause discomfort to the patient. In addition, local anesthesia is required for each procedure, making the overall treatment somewhat burdensome for patients. Therefore, I recommend enucleation rather than fenestration as the preferred surgical method. The authors should also provide a clear explanation of the rationale for selecting fenestration surgery in this study.
2, In general, Carnoy’s solution is known to have cytotoxic effects that may impair wound healing and potentially damage the peripheral nervous system. The solution contains components such as chloroform, ethanol, and glacial acetic acid, which exert strong tissue-fixative and necrotizing properties. While these effects are advantageous for eliminating residual cystic epithelial remnants and reducing recurrence rates, they can simultaneously delay tissue regeneration and compromise the healing process, particularly in areas adjacent to vital anatomical structures.
Moreover, previous reports have documented cases of temporary or even permanent sensory disturbances when Carnoy’s solution was applied near neural pathways, such as the inferior alveolar or mental nerve. For these reasons, its use requires careful consideration of both concentration and exposure time, as well as strict protection of surrounding soft tissues.
The authors should therefore provide a more detailed rationale for using Carnoy’s solution in their protocol, including the concentration employed, duration of application, and specific precautions taken to minimize neurotoxic or wound-healing complications. A discussion of alternative agents or modified formulations (e.g., chloroform-free Carnoy’s solution) would also enhance the scientific and clinical balance of this section.
Author Response
Dear Reviewer,
Thank you very much for your valuable review and for taking the time to provide insightful comments on our manuscript.
In response to your questions and remarks:
- Fenestration procedure
The fenestration was not performed for the purpose of decompression of CGCG (the lesion has a solid, not cystic structure), but rather to create an easy and precise access point for intralesional needle insertion. The windows created were very small, less than 0.5 cm² - allowing the surgeon to reliably identify the site and insert the needle without repeatedly “pecking” at the bone with its tip. In fact, procedures such as microsurgery of periapical tissues for chronic periodontitis or apicoectomy often require creating even larger bony windows.
All patients reacted very positively to the possibility of undergoing a less invasive, although longer and more burdensome, treatment method, and were willing to accept this approach. Even the pediatric patient treated with this method did not complain about the injections, which are comparable to routine local anesthesia used in dental procedures. The only concern raised by some patients during follow-up conversations was the inconvenience of traveling to the hospital for repeated visits, as several of them lived outside the city where our center is located.
2. Carnoy’s solution
We have expanded the discussion on Carnoy’s solution as suggested. Carnoy’s solution has been routinely used in our department for chemical curettage after the removal of odontogenic tumors, and we have not observed healing complications. The detailed composition of the solution is provided in the “Materials and Methods” section and in the caption of “Fig. 1.” According to our protocol, the exposure time of the post-CGCG cavity was 3 minutes, which is the same as recommended for other jawbone lesions. In the “Materials and Methods” section, we also emphasized the importance of protecting soft tissues and thoroughly irrigating any residual solution from the surgical cavity.
For these reasons, our preferred method of applying Carnoy’s solution is using a gauze wick soaked in the solution (as shown in two intraoperative photographs). This allows precise control over the area exposed to the solution, in contrast to, for example, syringe irrigation into the surgical site. The logistical challenges and potential risks associated with cryoablation of intraosseous lesions with liquid nitrogen (such as pathological fractures) are the main reasons why we have no clinical experience with this alternative method of adjunctive treatment.
Round 2
Reviewer 2 Report
Comments and Suggestions for Authors
hello
thank you for improving the paper
paper is more suitable in current form
thank you